# A Plant-Based Meal Stimulates Incretin and Insulin Secretion More Than an Energy- and Macronutrient-Matched Standard Meal in Type 2 Diabetes: A Randomized Crossover Study

**DOI:** 10.3390/nu11030486

**Published:** 2019-02-26

**Authors:** Hana Kahleova, Andrea Tura, Marta Klementova, Lenka Thieme, Martin Haluzik, Renata Pavlovicova, Martin Hill, Terezie Pelikanova

**Affiliations:** 1Institute for Clinical and Experimental Medicine, 14021 Prague, Czech Republic; KMarta@seznam.cz (M.K.); belenka@volny.cz (L.T.); halm@ikem.cz (M.H.); renata.pavlovicova@ikem.cz (R.P.); tepe@ikem.cz (T.P.); 2Physicians Committee for Responsible Medicine, Washington, 5100 Wisconsin Ave, NW, Suite 400, Washington, DC 20016, USA; 3Metabolic Unit, CNR Institute of Neuroscience, 35127 Padua, Italy; andrea.tura@cnr.it; 4Institute of Endocrinology, 11394 Prague, Czech Republic; mhill@endo.cz

**Keywords:** beta-cell function, incretins, insulin resistance, nutrition, plant-based, type 2 diabetes

## Abstract

Diminished postprandial secretion of incretins and insulin represents one of the key pathophysiological mechanisms behind type 2 diabetes (T2D). We tested the effects of two energy- and macronutrient-matched meals: A standard meat (M-meal) and a vegan (V-meal) on postprandial incretin and insulin secretion in participants with T2D. A randomized crossover design was used in 20 participants with T2D. Plasma concentrations of glucose, insulin, C-peptide, glucagon-like peptide-1 (GLP-1), amylin, and gastric inhibitory peptide (GIP) were determined at 0, 30, 60, 120, and 180 min. Beta-cell function was assessed with a mathematical model, using C-peptide deconvolution. Repeated-measures ANOVA was used for statistical analysis. Postprandial plasma glucose responses were similar after both test meals (*p* = 0.64). An increase in the stimulated secretion of insulin (by 30.5%; 95% CI 21.2 to 40.7%; *p* < 0.001), C-peptide (by 7.1%; 95% CI 4.1 to 9.9%; *p* < 0.001), and amylin (by 15.7%; 95% CI 11.8 to 19.7%; *p* < 0.001) was observed following consumption of the V-meal. An increase in stimulated secretion of GLP-1 (by 19.2%; 95% CI 12.4 to 26.7%; *p* < 0.001) and a decrease in GIP (by −9.4%; 95% CI −17.3 to −0.7%; *p* = 0.02) were observed after the V-meal. Several parameters of beta-cell function increased after the V-meal, particularly insulin secretion at a fixed glucose value 5 mmol/L, rate sensitivity, and the potentiation factor. Our results showed an increase in postprandial incretin and insulin secretion, after consumption of a V-meal, suggesting a therapeutic potential of plant-based meals for improving beta-cell function in T2D.

## 1. Introduction

The role of diet in the development of glucose intolerance and progression to type 2 diabetes (T2D) has been studied intensively [1,2,3]. Gastrointestinal hormones play a key role in glucose metabolism, energy homeostasis, satiety, and regulation of body weight [4]. Incretin hormones, namely glucagon-like peptide -1 (GLP-1) and gastric inhibitory peptide (GIP), which are released from the small intestine in response to nutrient ingestion to enhance glucose-dependent insulin secretion, aid in the overall maintenance of glucose homeostasis [5]. Furthermore, GLP-1 [6], and amylin [7] play a role in energy intake through appetite regulation. The release of these satiety hormones depends on meal composition. It has been suggested that particularly processed meat consumption leads to impaired release of gastrointestinal hormones [8].

Prospective observational studies indicate a positive relationship between a high consumption of red meat and T2D incidence [9,10]. This relationship is especially strong for processed meat. People who eat any processed meats are one third more likely to develop diabetes compared with those who do not consume any [11]. Several studies have pointed out a harmful effect of saturated fat on insulin resistance, glycemic control [1,12], and cardiovascular disease [13]. In contrast, vegetarians are about half as likely to develop diabetes, compared with non-vegetarians [14]. Randomized clinical trials showed a greater improvement in insulin sensitivity and glycemic control with a plant-based diet when compared with a conventional diet in participants with T2D [15,16].

We have investigated postprandial incretin and insulin secretion after ingestion of two meals matched for energy and macronutrient content: a standard meat burger and a plant-based burger, in men with T2D. Our hypothesis is that a plant-based meal stimulates the postprandial incretin and insulin secretion. Our results provide insight into the pathophysiology of T2D and potentially help build the evidence base for dietary recommendations for people with T2D.

## 2. Materials and Methods

### 2.1. Trial Design

This is a randomized crossover study of 20 men diagnosed with T2D, with two interventions. Written informed consent was obtained from all participants prior to enrollment in the study. This study was approved by the Ethics Committee of the Thomayer Hospital and Institute for Clinical and Experimental Medicine in Prague, Czech Republic on August 13, 2014. The protocol identification number is G14-08-42. The trial is prospectively registered with ClinicalTrials.gov (ID: NCT02474147).

### 2.2. Participants and Eligibility Criteria

Participants were men diagnosed with T2D aged between 30 and 65 years with a body mass index between 25 and 45 kg/m^2^; treated by lifestyle alone or with oral hypoglycemic agents (metformin and/or sulfonylureas) for at least one year, who had an HbA1c from ≥42 to ≤105 mmol/mol, (≥6.0 to ≤11.8%) and at least three symptoms of the metabolic syndrome. Exclusion criteria were renal, liver or thyroid disease, drug or alcohol abuse, unstable drug therapy, or a significant weight loss of more than 5% of body weight in the last three months.

### 2.3. Randomization and Masking

Participants were randomly assigned in a 1:1 ratio to start with the vegan meal (V-meal) or a meat sandwich (M-meal) based on a computer-generated randomization protocol by a study nurse. The randomization protocol could not be accessed beforehand. The interventions were unmasked. 

### 2.4. Interventions

The participants fasted overnight for at least 10 to 12 h and they did not take any of their diabetes medication the evening or the morning before the assessments. The meal consisted of either a processed meat burger (M-meal), or a plant-based tofu burger (V-meal). Meals were delivered fresh from the manufacturer. Both meals were served with a hot drink: The M-meal with café latte and the V-meal with green tea. Tap water was allowed ad libitum. Plasma concentrations of glucose, immunoreactive insulin, C-peptide, incretins, and amylin were measured at baseline, at 30, 60, 120, and 180 min after the meals. The participants came to our laboratory in the morning and, not knowing the sequence of the interventions, they were assigned to start with one of the test meals according to the randomization protocol. After a washout period of 1 week, the participants came back and completed the second meal test.

### 2.5. Measurements

The primary outcome was the area under the curve (AUC) for postprandial insulin secretion. The secondary outcome was the AUC for postprandial GLP-1 secretion. Tertiary outcomes included insulin resistance and beta-cell function.

#### 2.5.1. Anthropometric Measures and Blood Pressure

Height and weight were measured using a stadiometer and a periodically calibrated scale accurate to 0.1 kg, respectively. Resting blood pressure was measured after participants had been in a seated position for 5 min using a digital M6 Comfort monitor (Omron, Kyoto, Japan) three times at 2-min intervals. A mean value was calculated from the last two measurements.

#### 2.5.2. Metabolic Parameters

Plasma glucose was determined with the glucose-oxidase method using a Beckman Analyzer (Beckman Instruments Inc., Fullerton, CA, USA). Serum immunoreactive insulin and C-peptide were analyzed by radioimmunoassay using Immunotech Insulin and C-Peptide IRMA kits (Immunotech, Prague, Czech Republic), and glycated hemoglobin was assessed by a Bio-Rad Haemoglobin A1c Column Test (Bio-Rad Laboratories GmbH, Munich, Germany).

#### 2.5.3. Gastrointestinal and Appetite Hormones

The concentrations of GLP-1 active, GIP, and amylin total were measured via multiplex immunoanalyses based on xMAP technology using a Milliplex MAP Human Metabolic Hormone Magnetic Bead Panel (HMHEMAG-34K) (Millipore, Billerica, MA, USA) and a Luminex 100 IS analyzer (Luminex Corporation, Austin, TX, USA). 

#### 2.5.4. Beta-cell Function and Insulin Resistance

Beta-cell function was assessed by a mathematical model [17,18,19], as follows: Insulin secretory rates were calculated from plasma C-peptide concentrations by deconvolution [17] and expressed per square meter of estimated body surface area. The dependence of insulin secretory rates on glucose concentrations was modeled separately for each participant and each diet. Insulin secretion consists of two components. The first represents the static dependence of insulin secretion on glucose concentration and is characterized by a dose-response function. Relevant parameters include insulin secretion at 5 mmol/L glucose (fasting glucose level) and mean slope in the glucose range. The dose response is modulated by a potentiation factor, quantified as its ratio between 160 and 180, and 0 and 20 min values. The second component represents a dynamic dependence of insulin secretion on the rate of change of glucose and is determined by the rate sensitivity [18,19]. The model parameters (i.e., the parameters of the dose response, and the potentiation factor) were estimated from glucose and C-peptide concentration by regularized least squares [17,18,19]. Regularization involves the choice of smoothing factors that were selected to obtain glucose and C-peptide model residuals with standard deviations close to the expected measurement error (~1% for glucose and ~5% for C-peptide). Plasma insulin has been used for the assessment of insulin sensitivity, both in fasting and stimulated conditions. Estimation of the individual model parameters was performed by an investigator masked to group assignment. 

Insulin resistance was calculated by the Homeostasis Model Assessment (HOMA-IR) index [20]. In addition, PREDIM index (Predicted clamp-derived insulin sensitivity index from a standard meal test) was calculated as a measure of dynamic postprandial insulin sensitivity. It has been previously validated against clamp-derived measures of insulin sensitivity [21]. 

### 2.6. Statistical Analysis

We estimated the sample size using a power calculation with an alpha of 0.05 and a power of 0.80 to detect differences between the test meals in AUC for postprandial insulin secretion. This required 14 participants to complete both interventions. Using the trapezoid rule based on five time points (0, 30, 60, 120, and 180 min), we calculated the AUC for each plasma marker. We used repeated-measures ANOVA to analyze the data. Spearman’s correlations were calculated to test the relationship between postprandial changes in gastrointestinal hormones and changes in insulin secretion. They were calculated for the fasting plus for changes (30 to 0, 60 to 30, 120 to 60, and 180 to 120 min after ingestion of standard breakfast): For each period separately, and then for all 5 values combined. Analyses were undertaken using PASS 2005 statistical software (Number Cruncher Statistical Systems, Kaysville, UT, USA), with the statistician blinded to the analyses. All results are presented as means with 95% confidence intervals (CI).

## 3. Results

The flow of participants through the study is shown in Appendix A. The participant characteristics are shown in Table 1. The composition of the meals is shown in Table 2.

### 3.1. Postprandial Glucose and Insulin Response

Postprandial plasma glucose responses were similar after both test meals (*p* = 0.64; Figure 1A). An increase in the stimulated secretion of immunoreactive insulin (by 30.5%; 95% CI 21.2 to 40.7%; *p* < 0.001; Figure 1B), C-peptide (by 7.1%; 95% CI 4.1 to 9.9%; *p* < 0.001; Figure 1C), and amylin (by 15.7%; 95% CI 11.8 to 19.7%; *p* < 0.001; Figure 1D) was observed following consumption of the V-meal. 

### 3.2. Incretins

An increase in stimulated secretion of GLP-1 was evident (by 19.2%; 95% CI 12.4 to 26.7%; *p* < 0.001; Figure 1E) after the V-meal. A decrease in postprandial concentrations of GIP (by −9.4%; 95% CI −17.3 to −0.7%; *p* = 0.02; Figure 1F) was observed after the V-meal. The detailed time flow of postprandial concentrations of glucose, insulin, and incretins, are shown in Appendix A.

### 3.3. Beta-cell Function and Insulin Resistance

Parameters of beta-cell function and insulin resistance are shown in Table 3. Insulin secretion at a fixed glucose value 5 mmol/L was higher after the V-meal by 11.4% (95% CI 1.6 to 22.4%; *p* = 0.04). Rate sensitivity was increased after the V-meal by 55% (95% CI 3.0 to 132.1%; *p* = 0.04), and so was the potentiation factor by 13.3% (95% CI 6.7 to 20.0%; *p* = 0.02). No differences between the meals were observed in glucose sensitivity or insulin resistance assessed by HOMA-IR and PREDIM. 

### 3.4. Correlations of Changes in Gastrointestinal Hormones with Glucose Metabolism

Postprandial secretion of GLP-1 and amylin increased in parallel with C-peptide concentrations in each participant group in time span from 0 to 180 min after the meal ingestion, as shown in Appendix A. A positive relationship was found between Δ GLP-1 and Δ C-peptide (*r* = 0.416, *p* < 0.001), and between Δ amylin and Δ C-peptide (*r* = 0.829, *p* < 0.001).

## 4. Discussion

Our study showed different postprandial incretin and insulin secretion following the ingestion of a V-meal when compared with a standard M-meal, matched for energy and macronutrient composition. Overall postprandial plasma glucose responses did not differ between meals, and we detected significantly higher stimulated insulin and C-peptide responses, higher concentrations of GLP-1, and lower levels of GIP, after the plant-based meal compared with the standard M-meal. Several parameters of beta-cell function were improved after the plant-based meal, namely total insulin secretion, insulin secretion at a fixed glucose value 5 mmol/L, rate sensitivity, and the potentiation factor. No difference was observed in HOMA-IR or PREDIM as measures of insulin resistance.

### 4.1. Insulin Secretion and Beta-cell Function

That a single plant-based meal can increase postprandial insulin secretion has direct implication for diabetes treatment. Preserving the capacity of beta-cells to produce insulin according to changing need is a cornerstone in the treatment of diabetes [22]. Insulin secretion and beta-cell function may be improved by different treatment options that lower body fat (such as diet and exercise, GLP-1 agonists, or bariatric surgery) or change fat distribution (such as thiazolidinediones) [22,23]. As medications and bariatric surgery are expensive and introduce potential side effects, lifestyle interventions should be the first-choice treatment. It has been demonstrated that a 16-week vegan diet improves insulin resistance and beta-cell function in overweight individuals, addressing both core pathophysiologic mechanisms involved in diabetes at the same time [24].

### 4.2. Incretins

We observed an increase in postprandial GLP-1 concentrations and a decrease in GIP levels after the V-meal. In addition, there was a strong positive relationship between the changes of GLP-1 and changes in C-peptide secretion. The incretin release accounts for 50% to 70% of total postprandial insulin secretion. GLP-1 has been primarily associated, together with GIP, with the incretin effect, the postprandial augmentation of insulin secretion by gut hormones [25]. It is well understood that in patients with diabetes, the incretin effect is diminished [26,27] due to impaired beta-cell sensitivity [28]. Several hypotheses have been formulated to explain loss of beta-cell sensitivity. Widely accepted concepts include hyperglycemia- and hyperlipidemia-associated receptor desensitization [29].

We have demonstrated previously that a processed M-meal leads to postprandial hyperlipidemia, increased lipoperoxidation, persistent postprandial hyperinsulinemia, and lower secretion of gastrointestinal hormones, including GLP-1, compared with an energy-matched V-meal in people with T2D [8]. Interestingly, in the previous study, V-meal resulted in higher GLP-1 and GIP secretion. It has been shown that GIP secretion may also reflect the quality of carbohydrates [30], which may have influenced the findings. The current study used cooked meat in the standard M-meal and we matched both meals in macronutrient content in order to provide further insights into the pathophysiology of incretin and insulin secretion in T2D. It is also important to note that some dietary interventions, such as a whole grain diet, have been shown to improve beta-cell function independent of incretin secretion. In this particular study, GLP-1 secretion was lower and GIP remained unchanged after 8 weeks of a whole-grain diet compared with a refined-grain diet in people with prediabetes [31].

Several studies have suggested that a high intake of saturated fat naturally present in meat contributes to the risk of glucose intolerance [1,12] due to impaired beta-cell sensitivity and function. Fat quantity and quality have been shown to affect the gastrointestinal peptide release in people with metabolic syndrome [32].

In addition to fat quality, carbohydrate quality needs to be taken into account. The V-meal contained naturally more fiber and less sugar than the M-meal and these differences may have influenced the findings [32].

Another important mechanism is replacing animal protein with plant protein, which has been shown to reduce blood lipids and blood pressure [33], to lower fasting plasma glucose and fasting insulin levels, and to improve glycemic control in diabetes [34]. It is important to note that in order to see changes in fasting plasma glucose and insulin levels, a longer-term intervention is needed. As our study was an acute feeding trial, we were only able to track the differences in postprandial changes.

Additionally, plant-foods are naturally rich in fiber, which may have influenced postprandial insulin levels [35,36]. Furthermore, micronutrients, such as zinc [37], polyphenols [38], and other phytochemicals may also have played their positive role in postprandial incretin and insulin secretion. Our results are in line with these hypotheses and further demonstrate that a plant-based meal may improve the secretion of incretins and insulin in people with T2D where secretion has already been compromised.

The decreased postprandial plasma concentrations of GIP detected following ingestion of the V-meal compared with the M-meal are a further positive finding, as increased levels of the anabolic hormone GIP have been shown to directly promote fat storage and energy deposition [39]. While GLP-1 levels are usually diminished in T2D, GIP levels may be increased due to lowered beta-cell GIP-receptor expression, resulting in GIP resistance in T2D [40]. Interestingly, not only dietary interventions, but also metabolic surgery affects the secretion of incretins. For example, duodenal-jejunal bypass liner, and endoscopic method that mimics small intestine mechanisms of a Roux-en-Y gastric bypass, has also been shown to increase GLP-1 and decrease GIP concentrations [41]. Therefore, decreased GIP concentrations in our study may reflect lower GIP resistance after the plant-based meal.

A consensus report by the American Diabetes Association and the European Association for the Study of Diabetes recommends an incretin-based therapy (GLP-1 analogue or DPP-4 inhibitor) following the failure of metformin monotherapy [42]. Improvement in beta-cell function is considered one of the main benefits of incretin-based therapy [43]. Our data suggest that plant-based meals may also improve beta-cell function, which may have important clinical implications for dietary treatment of T2D.

### 4.3. Amylin

Amylin concentrations were higher after the V-meal, a result substantiated by the positive relationship between changes in amylin and changes in C-peptide. Amylin is understood to regulate both glucose and energy homeostasis [44] and is one of the prominent satiety signals, reducing eating by promoting satiation [45]. Therefore, the increase in postprandial concentrations of amylin after the V-meal observed in our study is a positive finding.

### 4.4. Strengths and Limitations

The main strength of this study is our comprehensive measurement of the postprandial state in a physiological setting, after ingestion of two sandwiches. We identified the mechanisms behind the improvements in glucose metabolism associated with plant-based diets. We recorded markers of both signal and response to better consider the interplay of digestion and metabolism. Finally, our meals were commonly consumed meals served in amounts corresponding to those ingested during a typical meal, making the study results highly applicable and practical. 

We also recognize several limitations. The diabetes duration was fairly short in our study participants, increasing the likelihood to observe an increase in postprandial incretin and insulin secretion. This observation may not be generalizable for people with T2D with a long duration of the disease. Furthermore, our study included two test meals, and did not reflect habitual dietary patterns. The differences in postprandial incretin and insulin secretion identified in this study suggest that longer-term studies would be beneficial to show if a vegan diet may prevent the development of diabetes or slow down diabetes progression. The meals differed in saturated fat, dietary fiber, sugar content, amino acid composition, micronutrient content, polyphenols, and other phytochemicals, all of which may have influenced our results. Additionally, although we used two hot caffeinated drinks with both meals, coffee and green tea may have influenced glucose and insulin secretion slightly differently [46]. Furthermore, we have not matched the meals in micronutrient content and have not tested the bioavailability of these micronutrients after digestion of both meals. However, these components are characteristic for plant-based meals compared with standard meals containing meat and other animal products, so while this difference is not controlled for in this study, it does increase the generalizability of our results.

## 5. Conclusions

In conclusion, our study results indicate an improvement in postprandial incretin and insulin secretion in people with T2D, following consumption of a V-meal, suggesting a therapeutic potential of plant-based meals for improving beta-cell function in T2D.

## Figures and Tables

**Figure 1 nutrients-11-00486-f001:**
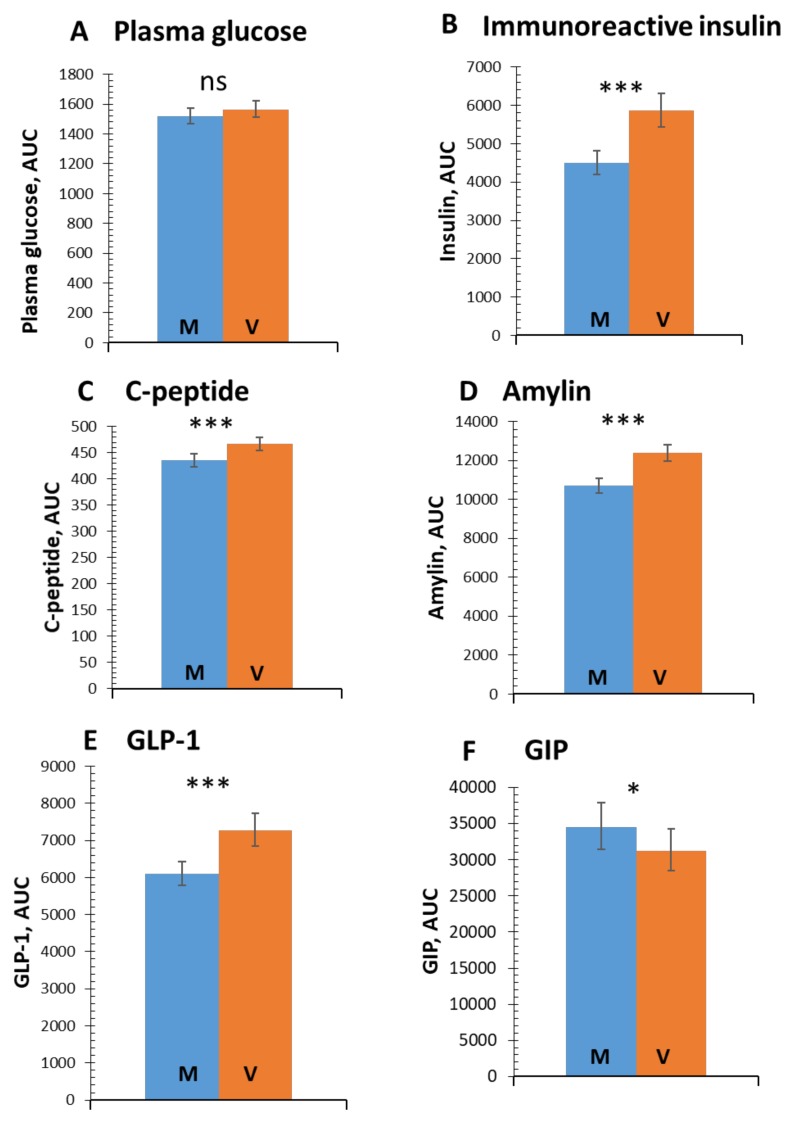
Area under the curve (AUC) for postprandial changes in plasma glucose, immunoreactive insulin, C-peptide, amylin, glucagon-like peptide -1 (GLP-1) and gastric inhibitory peptide (GIP), after the ingestions of the standard meat (M-meal) (blue bars) and the energy- and macronutrient-matched vegan (V-meal) (orange bars). Data are expressed as mean with 95% confidence intervals (CI), using a repeated-measures ANOVA. *P*-values are marked as ns (nonsignificant) for *p* ≥ 0.5, * *p* < 0.05, and *** *p* < 0.001.

**Table 1 nutrients-11-00486-t001:** General characteristics of the diabetic population.

Characteristic	Patients with T2D (*n* = 20)
Age (years)	47.8 ± 8.2
Weight (kg)	108.2 ± 11.9
Body mass index (kg∙m^−2^)	34.5 ± 3.4
Waist circumference (cm)	106.9 ± 23.6
HbA1c (IFCC; mmol/mol)	48.5 ± 8.1
HbA1C (DCCT; %)	6.6 ± 1.0
Fasting plasma glucose (mmol/L)	7.2 ± 1.5
Triglycerides (mmol/L)	2.1 ± 1.1
Total cholesterol (mmol/L)	4.5 ± 0.9
LDL-cholesterol (mmol/L)	2.6 ± 0.1
HDL-cholesterol (mmol/L)	1.0 ± 0.2
Systolic blood pressure (mm Hg)	144.4 ± 13.4
Diastolic blood pressure (mm Hg)	96.2 ± 8.8
Duration of diabetes (years)	4.25 ± 3.25

Data are means ± SD.

**Table 2 nutrients-11-00486-t002:** Composition of the test meals together with the drinks.

Meal	M-meal	V-meal
Total weight (g)	200	200
Energy content (kCal)	513.6	514.9
Carbohydrates (g) (%)	55 (44.8%)	54.2 (44.0%)
Sugar (g) (%)	21 (17%)	4 (3%)
Proteins (g) (%)	20.5 (16.7%)	19.9 (16.2%)
Lipids (g) (%)	22 (38.6%)	22.8 (39.8%)
Saturated fatty acids (g)	8.6	2.2
Fiber (g)	2.2	7.8

The postprandial state was measured after intake of a standard breakfast—one of two energy- (514 kcal) and macronutrient-matched meals (45% carbohydrates, 16% protein, and 39% lipids) in a random order: Either a standard meat burger (M-meal; cooked-pork seasoned meat in a wheat bun, tomato, cheddar-type cheese, lettuce, spicy sauce) together with 300 mL café latte with 21 g sugar, or a plant-based burger (V-meal; tofu burger with spices, ketchup, mustard, tomato, lettuce and cucumber in a wheat bun) together with 300 mL of unsweetened green tea.

**Table 3 nutrients-11-00486-t003:** Beta-cell function and insulin resistance after ingestion of a M-meal and a V-meal.

	M-meal	V-meal	*p*-value
Insulin secretion/ Beta-cell function			
Basal insulin secretion (pmol min^−1^ m^−2^)	151.9 (146.6–157.2)	153.9 (148.6–159.3)	0.72
Total insulin secretion (nmol m^−2^)	62.2 (60.2–64.3)	68.2 (66.1–70.4)	<0.001
Insulin secretion at a fixed glucose value (5 mM) (pmol min^−1^ m^−2^)	99.0 (84.3–114.3)	110.3 (95.2–126.1)	0.04
Insulin secretion at a fixed glucose value (6 mM) (pmol min^−1^ m^−2^)	166.2 (146.3–186.8)	174.6 (154.5–195.6)	0.08
Insulin secretion at a fixed glucose value (7 mM) (pmol min^−1^ m^−2^)	234.3 (209.9–260.3)	234.0 (209.7–260.1)	0.89
Glucose sensitivity (pmol min^−1^ m^−2^ mM^−1^)	75.3 (65.7–85.8)	72.0 (62.6–82.2)	0.33
Rate sensitivity (pmol m^−2^ mM^−1^)	202.4 (100.5–300.1)	313.7 (167.7–448.6)	0.04
Potentiation factor ratio (dimensionless)	1.5 (1.4–1.6)	1.7 (1.6–1.8)	0.02
Insulin sensitivity/resistance			
HOMA-IR (dimensionless)	4.2 (3.9–4.5)	4.2 (4.0–4.5)	0.90
PREDIM (mg min^−1^ kg^−1^)	2.7 (2.6–2.8)	2.6 (2.5–2.7)	0.73

Data are expressed as means with 95% confidence intervals. Listed *p* values are for the difference between the meals assessed by repeated measures ANOVA.

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
