# Peer review of "A Plant-Based Meal Stimulates Incretin and Insulin Secretion More Than an Energy- and Macronutrient-Matched Standard Meal in Type 2 Diabetes: A Randomized Crossover Study"

_nutrients, 2019, doi:10.3390/nu11030486_

Reviewer 1 Report

Please see my comments below:

1) abstract: please provide a sentence describing the background of the study.

2) the age and BMI range were very wide. How does this influence the results? Especially the BMI range, ranging from overweight to extremely obese.

3) Please describe this in detail: a random sequence generator

4) Why tap water was used?

5) for the results, please correct for the confounders.

6) Table 2: how about the presence of micronutrients suc as zinc that might have a modulation effect on diabetes management? Please consider this as well

7) Figure 1: how long is the washout period?

8) Please decribe how the cross over was performed. This is not included in the methods section.

9) Figure 1: this is poorly done. Please modify according to the Consort statment.

10) What was the primary and secondary outcomes? Was the sample size calculated according to primary outcome? The authors poorly reported this.  

Author Response

We thank Reviewer 1 for the insightful comments. We have addressed all of them one-by-one. The changes in the manuscript are highlighted in yellow. Thank you for your help in improving our paper.

Reviewer 2 Report

Kahleova H et al. compared the metabolic effects of standard meat (M) meal with vegan (V) meal in patients with type 2 diabetes in term of postprandial hormone levels. The study is interesting however several concerns are raised.

Major points

In addition to the different M and V meal      consumption, the participants drank Café Latte with 21 g sugar in the M      group however participants in the V group consumed unsweetened green tea.      Since sugar is known to change glucose, insulin and other hormone levels      (Chang CY et al Eur J Nutr. 2018 Feb;57(1):179-190), the reported changes      in postprandial insulin, amylin and GLP-1 levels after the consumption of      V meal could be in part attributed to the presence or absence of sugar      found in the drink. This is a very important bias, which impacts the      conclusion of the data. What was the reason for using drinks with      different sugar content for the different meals?

Since there are many studies, which      investigated the effect of replacing animal protein with plant protein in      the diet, the authors should discuss more thoroughly their findings in the      view of the current literature. Furthermore, there are several meta-analysis      summarizing the current literature like Viguiliouk E et al Nutrients. 2015      Dec 1;7(12):9804-24 or Chalvon-Demersay T et al J Nutr. 2017      Mar;147(3):281-292. Many studies showed that replacing animal protein with      plant protein in the diet did not affect fasting insulin level, however      some studies suggest a lower fasting insulin level after the application      of plant protein based diets.

In a former study, the authors      investigated an isocaloric vegan meal compared to processed meat meal      without macronutrient matching using patients with type 2 diabetes Belinova      L et al PLoS One. 2014 Sep 15;9(9):e107561. The authors already observed      the vegan meal elevated postprandial gastrointestinal hormones compared to      meat meal, however this former study mainly contradicts the findings of      the current study. Although the former and the current study used      different vegan meals, the observed differences between the two studies should      be discussed.

Furthermore, the M meal contains higher      levels of saturated fatty acids and lower levels of fibers compared to V      meal. The authors shortly mention these differences in the discussion but      saturated fatty acids and fibers could indeed change postprandial hormone      levels, therefore they needed to be discussed in more details. In order to      determine whether the observed differences at the hormone levels of T2D      patients are attributed to the fatty acids, one could design a study using      for eg. the vegan meal and compare it with the “same” vegan meal      complemented with high saturated fatty acids. Diets rich in unsaturated      fatty acids (MUFA or PUFA diet) showed significantly lower level of GIP in      patients with metabolic syndrome compared to saturated fatty acid rich      diet (Chang CY et al Eur J Nutr. 2018 Feb;57(1):179-190). In addition to      saturated fatty acids, fibers were also showed to impact glucose      metabolism (Post RE et al J Am Board Fam Med. 2012;25:16–23).

Minor points

The decreased GIP level upon V meal is not      discussed in the manuscript.

The time period between the two different      meal interventions is not indicated.

Author Response

We thank Reviewer 2 for the insightful comments. We have addressed all of them one-by-one. The changes in the manuscript are highlighted in yellow. Thank you for your help in improving our paper.

Round  2

Reviewer 2 Report

Kahleova et al revised their manuscript, which is improved. However several points remain still unclear.

According      to the first answer of authors, table 2 depicts the composition of the      test meals in combination with the used drinks. This fact is still not      well indicated, therefore it should be better pointed out in the title of      table 2 and also in the methods section. As mentioned earlier, sugar      consumption directly influences glucose and hormonal metabolism, therefore      sugar content (in %) should be indicated for both meals in table 2. Did      the authors use the same amount of sugar for both meals? If not, than      different sugar content is likely another yet unconsidered factor, which      probably impacts the observed findings and therefore need to be discussed.      Furthermore, using Café Latte in one study group and green tea in the      other was probably not a good idea, since studies showed the green tea could      impact glucose metabolism, however coffee did not (for review see Kondo Y      et al Nutrients. 2018 Dec 27;11(1). These possible factors need to be also      discussed.

According      to the observed findings and the literature, the authors now integrated      the proposed references, however they did not discuss the conflicting      findings. They only mention the studies, which fit to their data. The authors      need to critically discuss their findings in the view of the current      literature and should find possible explanation for the discrepancy      between the observed findings in this manuscript and findings from other      studies. For eg why other studies reported lower fasting insulin levels, which      contradicts Suppl. Fig. 2B? Why does the former study of the authors Belinova      et al Plos One 2014 contradict the findings of the current manuscript (like      decreased GLP1 in the former and increased GLP1 in the current study)?

The      decreased level of GIP upon V meal treatment is now shortly mentioned as “positive      finding”, however it contradicts the “incretin effect” on postprandial      insulin secretion discussed in line 211-217. If V meal increased GLP1      level, why did it decrease GIP?

Minor      corrections are needed like “8,2” (in table 1 should be corrected as 8.2),      “HOMA” or “HOMA-R” (correct to HOMA-IR), typos like: “stadied”, “insuloin”.      In line 82, “4. Interventions” need to be moved in a new row. Affiliation      is missing for the last author.

Author Response

We thank reviewer 2 for the insightful comments and all the effort to help us improve our manuscript. We highly value all the inputs and have incorporated them into our manuscript.
